# How to Train Your Diffusion Models for Few-Shot Classification

## Abstract

Generation and classification are two sides of the same coin: a strong generative model can be transformed into a powerful classifier. This is evident when diffusion models (DMs) outperform CLIP-based approaches in fine-grained or customized classification tasks, where a small few-shot training set defines the task on the fly. In this setting, the model is typically fine-tuned to reconstruct the training samples and, at inference, predicts the label with the lowest expected reconstruction error across diffusion time-steps. Although effective, this approach is computationally expensive, as it requires computing *average* reconstruction errors for *every class* over the *full range of time-steps* and *multiple sampled noises*. In this work, we study techniques to improve both efficiency and accuracy of diffusion classifiers. To accelerate inference, we propose dynamic time-step selection to minimize unnecessary evaluations. To improve the estimation of reconstruction errors, we introduce class-object mask learning, which reduce variance and thereby require fewer noise samples to achieve high precision. To further reduce the number of candidate classes, we explore candidate class selection. Together, these techniques speed up diffusion-based classifiers by over an order of magnitude while simultaneously maintaining or even improving classification performance. Finally, we show that DMs and CLIP-based models are complementary, and integrating the two achieves further gains — reinforcing the close connection between generation and classification.

## 1 Introduction

While multi-modal foundation models(eg. Chen et al. (2024), Achiam et al. (2023)) excel at general tasks, they often struggle in zero-shot scenarios requiring fine-grained or customized recognition. This gap highlights the need for few-shot learning, which adapts these powerful models to novel concepts defined by a small support set (Liu et al., 2024). In few-shot classification, two dominant strategies have emerged. The first adapts pre-trained discriminative models like CLIP (Radford et al., 2021b), directly aligning image features with text embeddings. The second, known as Diffusion Classifiers (Li et al., 2023; Clark & Jaini, 2023), repurposes generative diffusion models for classification by determining which class label offers the best guidance for denoising an image. Extensive experiments show that diffusion classifiers can outperform CLIP-based methods, underscoring the promise of generative adaptation for few-shot tasks.

The classification principle of a diffusion classifier is formalized in Equation 1 and 2 below[1].

$$E(x, t, c) := Square(\mathrm{DM}(\, t\mathbf{x} + (1-t)\epsilon, t, c) - \epsilon) \in R^{Ch \times H \times W}, \epsilon \; is \; inner \; sampled \quad (1)$$

$$\hat{c} = \overbrace{\underset{c \in \{1,...,K\}}{\arg\min}}^{\text{classes}} \underbrace{\mathbb{E}_{\epsilon \sim \mathcal{N}(\mathbf{0},\mathbf{I})}}_{\text{noises}} \overbrace{\mathbb{E}_{t \sim U(0,1)}}^{\text{time-steps}} \|E(x, t, c)\|. \quad (2)$$

---

[1]For simplicity, we present noising process using flow matching (Lipman et al., 2022), while most existing diffusion classifiers are built on DDPM(Ho et al., 2020). For DDPM, the time-step range is often defined as $[0, 1000]$ rather than $[0, 1]$.

A diffusion model (DM) learns to predict the noise $\epsilon$ added to an image $\mathbf{x}$ at a given time-step $t$, conditioned by class $c$. To classify an image, the model identifies the target class $\hat{c}$ whose class condition allows diffusion to most accurately reconstruct the original image by predicting the added noise. This is achieved by finding the class that minimizes the expected error, referred to as **reconstruction error**(the L2-Norm of reconstruction error map $E(\mathbf{x}, t, c)$), between the predicted noise and the ground-truth noise $\epsilon$. As the formula shows, this process requires averaging the result for *every class* over *many randomly sampled noises and time-steps* to achieve a stable prediction. Consequently, classifying a single image demands numerous forward passes through the network, making this method computationally expensive and prohibitively slow for practical applications. Though strategies like compulsory sampling timesteps (Yoon et al.), noise learning (Wang & Chen, 2025), and maximum confidence filtering (Li et al., 2023) have been proposed to address this, they either lead to a notable drop in accuracy or lack the efficiency required for practical application.

In this work, we introduce FastTiF, a novel method that significantly accelerates TiF (Yue et al., 2024), the best performing Diffusion classifier for few-shot learning framework, while maintaining or even enhancing its performance by addressing its three primary computational bottlenecks mentioned earlier: extensive time-step sampling, uniform noise evaluation, and a large pool of class candidates. First, regarding **time-steps**, we find that a diffusion classifier's discriminative ability peaks within a narrow, optimal **time-step range**. FastTiF identifies this range from training data and concentrates sampling within it, drastically reducing redundant computations. Second, for **noise evaluation**, we posit that reconstruction errors in background regions are less informative for classification. We therefore introduce a **spatial mask**, generated by a segmentation model finetuned to predict ground-truth masks that are first derived from the reconstruction error maps of the training data, allowing it to selectively down-weight these irrelevant areas and focus on salient object features. Finally, to streamline the selection of **class candidates**, FastTiF leverages the powerful cross-grained discrimination ability of a CLIP-based method to **filter** the list of potential class candidites for each sample. This not only reduces the number of required inferences but can also improve accuracy by eliminating unlikely candidates. Collectively, these three optimizations allow FastTiF to achieve substantial speed improvements while *maintaining or even enhancing classification performance*.

Our experimental results indicate that a diffusion classifier generally surpasses the performance of most CLIP-based methods, particularly in fine-grained and customized classification tasks, while maintaining a computational speed suitable for practical applications. It should be noted, however, that certain CLIP-based methods can achieve superior results. To further elucidate the sources of our model's efficacy, we conducted a rigorous analysis of the specific impact of several key components—namely, class candidate filtering, mask learning, and timestep selection—on both the overall performance and computational speed of the diffusion classifier. Finally, we discuss the potential for a synergistic approach that leverages the respective strengths of both models. Specifically, CLIP's proficiency in rapid inference and cross-grained discrimination can be utilized to filter class candidates, while the diffusion classifier can resolve the ambiguities encountered by CLIP in fine-grained classification tasks.

To summarize, our three main contributions are:

- We propose FastTiF, a novel method that significantly accelerates the performance of diffusion-based few-shot classification while maintaining or even improving performance.

- We introduce time-step learning, mask learning and class candidate filtering mechanism to improve the efficiency and discriminative power of the diffusion classifier. And further discuss how key components affect performance and acceleration.

- We present the possibility to leverage the strengths of both discriminative and generative models, and demonstrate a promising path for accelerating diffusion-based classifiers.

## 2 RELATED WORK

**Conventional FSL** typically centers on adapting the generalized representational ability of a pretrained model—an ability bestowed by the rich, diverse knowledge accumulated during its large-scale pretraining phase—to specific downstream domains, and this adaptation is predominantly achieved through finetuning (Chen et al., 2019; Nakamura & Harada, 2019). The pretraining stage

of this paradigm is foundational: here, the model is trained on extensive pretraining datasets to learn robust, transferable representations of the data, encoding features and concepts that lay the groundwork for subsequent few-shot task adaptation (Wang et al., 2020). In the finetuning stage, which is tailored to the data-scarce few-shot setting, the framework usually leverages the small-sized support set (containing just a few labeled samples per novel class) to define and model the downstream concepts (Snell et al., 2017). After adjusting the model with the support set, it then conducts inference on the query set (samples from the same novel classes as the support set) to assess performance.

**FSL with Foundation Models** The preliminary alignment of visual and textual features achieved by CLIP (Radford et al., 2021b) has opened new avenues for few-shot learning (FSL). The primary challenge in adapting CLIP for FSL tasks is to further refine this alignment within specific, fine-grained domains. Previous works have addressed this challenge from three distinct perspectives. The first is prompt tuning, whose objective is to learn a specific prompt for each class: CoOp (Zhou et al., 2022a) develops a continuous prompt embedding rather than relying on a manually designed prompt, CoCoOp (Zhou et al., 2022b) builds on CoOp by learning a prompt conditioned on the input image, ProGrad (Zhu et al., 2023) adjusts the prompt gradient to align with CLIP's general knowledge, and MaPLe (Khattak et al., 2023) goes a step further by also fine-tuning CLIP's visual encoder. The second category focuses on learning an adapter for CLIP's visual features: CLIP-Adapter (Gao et al., 2024) employs a lightweight residual-style adapter, followed by the training-free Tip-Adapter (Zhang et al., 2021), CALIP (Guo et al., 2023) introduces a parameter-free attention mechanism to enhance both zero-shot and few-shot performance, CaFo (Zhang et al., 2023) combines multiple foundation models to facilitate feature adaptation. The third category involves leveraging multiple modalities and specialized optimization strategies that move beyond simple prompt or adapter tuning: SADA (Wang et al., 2023) applies prompt tuning, visual attacking module and trainable visual-language prototype for further alignment; Multi-modality (Lin et al., 2023) finetunes both visual and textual encoders with a mechanism similar to contrastive learning but more tailored for few-shot learning; AMU-Tuning (Tang et al., 2024) integrates knowledge from auxiliary vision models with CLIP's output via optimizable modules. These methods represent a shift towards more sophisticated, multi-component frameworks for adapting large foundation models.

## 3 SETTINGS

**Few-shot Classification**. We formally address the problem of $K$-way-$N$-shot few-shot classification, a paradigm designed to evaluate a model's ability to learn from a minimal number of examples. In this setting, the task is structured into distinct learning episodes, each comprising a support set $S = \{(x_i, y_i)\}_{i=1}^{K \times N}$ and a query set $Q$. The support set serves to define the specific classification problem for the episode, containing a total of $K \times N$ labeled examples, where there are $K$ distinct classes and exactly $N$ unique labeled samples, or "shots", for each of these classes. Commonly, these concepts and features rarely appear in the pretraining dataset of multimodal foundation models, making it hard for these models to perform well zero-shot. The fundamental objective is to leverage the sparse information within the support set S to train or adapt a model that can accurately predict the labels for new, previously unseen query examples drawn from the same $K$ categories. The model's performance on the query set thereby measures its capacity for rapid generalization and adaptation to the novel concepts established by the limited examples provided in the support set.

**FSL with CLIP-based Models**. A direct way for few-shot classification is to adapt CLIP-based models, which contains a pair of image encoder $V$ and text encoder $T$. Given image $\mathbf{x}$ and class text prompt $y_c$ of class $c$, the CLIP similarity is defined as Equation 3, where image and class text prompt are both encoded by their encoders into vectors of same dimensions and calculate cosine similarity. The target class $\hat{c}$ of $\mathbf{x}$ is expected to have the largest CLIP similarity given as Equation 4. The few-shot setting typically adapts image encoder $V$ and text encoder $T$, such that image features and texual features of the few-shot setting are aligned.

$$\text{CLIP}(\mathbf{x}, y_c) := \cos(V(\mathbf{x}), T(y_c)) \tag{3}$$

$$\hat{c} = \underset{c \in \{1, ..., K\}}{\arg\min} \text{CLIP}(\mathbf{x}, y_c) \tag{4}$$

**FSL with Diffusion Classifier**. Diffusion classifiers (Li et al., 2023; Clark & Jaini, 2023) adapt the generative power of diffusion models for classification tasks by fine-tuning pre-trained model

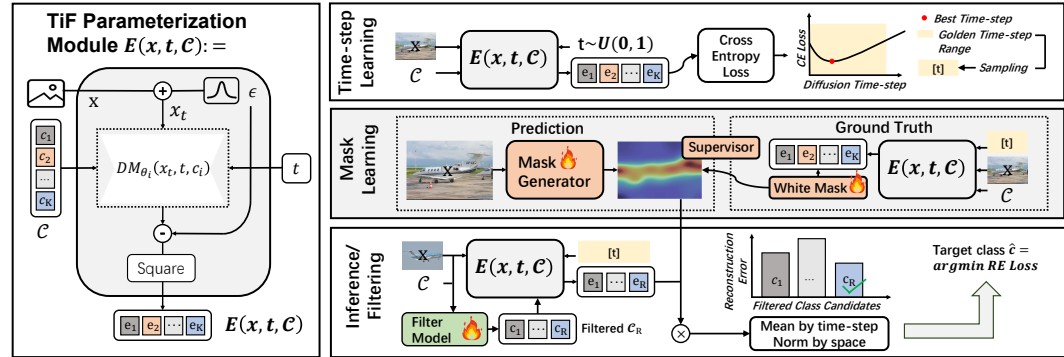

Figure 1: Overall Pipeline of FastTiF. The framework has three core components and one inference stage: (1) **TiF Parameterization Module**: Outputs class-wise reconstruction error maps $E(\mathbf{x}, t, C)$ ($E(\mathbf{x}, t, C) := Concat[E(\mathbf{x}, t, c) \ for \ c \in C]$) between noisy $\mathbf{x_t}$ and $DM_{\theta_i}(\mathbf{x_t}, t, c_i)$ predictions. (2) **Time-step Learning**: Uses $E$ to build cross-entropy loss for evaluation of diffusion classifier's discriminative ability. This ability peaks at a "Best Time-step" (red); sampling nearby (yellow) is optimal. (3) **Mask Learning**: Trains a generator to produce sample-specific masks for focusing on discriminative features. **During inference**: A fine-tuned filter reduces $C$ to $C_R$ ($C$ denotes the full class set). Masked $\mathbf{E}$ (at golden $t$ range) determines the final class.

for each candidate class c in a support set using a class-specific DreamBooth LoRA (Ruiz et al., 2023; Hu et al., 2022), denoted as $\theta_c$, to create a specialized network, $DM_{\theta_c}$. During the inference stage, an image $\mathbf{x}$ is classified by selecting the target class $\hat{c}$ whose corresponding class-conditional model $DM_{\theta_c}$ and class condition $c$ minimize the reconstruction loss when attempting to denoise the image. This process, however, is susceptible to learning spurious correlations from the training data, where the model might associate irrelevant features with a class. To address this, the TiF (Time-step Few-shot) framework introduces a specific ratio, $r_t$, which is applied during the process to de-bias the model's decision-making, effectively eliminating the influence of these spurious correlations and improving the robustness and accuracy of the classification (Yue et al., 2024).

## 4 APPROACH

The inefficiency of a standard diffusion classifier stems from its need to compute an average reconstruction error for every class candidate, across a wide range of diffusion time-steps and for multiple noise samples. In our approach, we systematically address these bottlenecks by introducing acceleration methods across these three key dimensions: the range of time-steps, the number of noise samples, and the set of class candidates. We plot our pipeline in Figure 1 and detail the rationale behind each method and analyze its impact on both inference speed and classification performance.

### 4.1 TIME-STEPS

As proposed in prior work like TiF, different diffusion time-steps correspond to loss of different granularity of image features(eg. "windows" attribute is lost at early time-steps, and "background" attribute is lost at later time-steps), thus not all time-steps contributes equally to classification (Yue et al., 2024). Inspired by this, we hypothesize that not all time-steps are equally discriminative for classification. We find that a "golden" range of time-steps exists where the model's ability to distinguish between classes is maximized through experiments. To leverage this, we propose a time-step learning mechanism.

**Time-step Learning**. Let $L_t(x, c) := \mathbb{E}_{\epsilon \sim \mathcal{N}(\mathbf{0}, \mathbf{I})} \|E(\mathbf{x}, t, c)\|$. For each training sample x, we compute and store the reconstruction error $L_t(\mathbf{x}, c)$ for every class candidate $c$ across a multitude of sampled time-steps $t$.

$$F_t := \underset{c \in \{1, \dots, K\}}{L_{CE}} (-L_t(\mathbf{x}, c)). \tag{5}$$

We then formalize the classifier's discriminative ability at a specific time-step $t$ using the cross-entropy loss over the negative reconstruction errors, as defined in Equation 5, it can be regarded as a **negative correlated metric** of discrimination ability. The optimal time-step $t^*$ for a given sample $\mathbf{x}$ for discrimination is therefore the one that **minimizes** this value, as shown in Equation 6. During inference, we sample time-steps from a narrow distribution centered around the average $t^*$ value

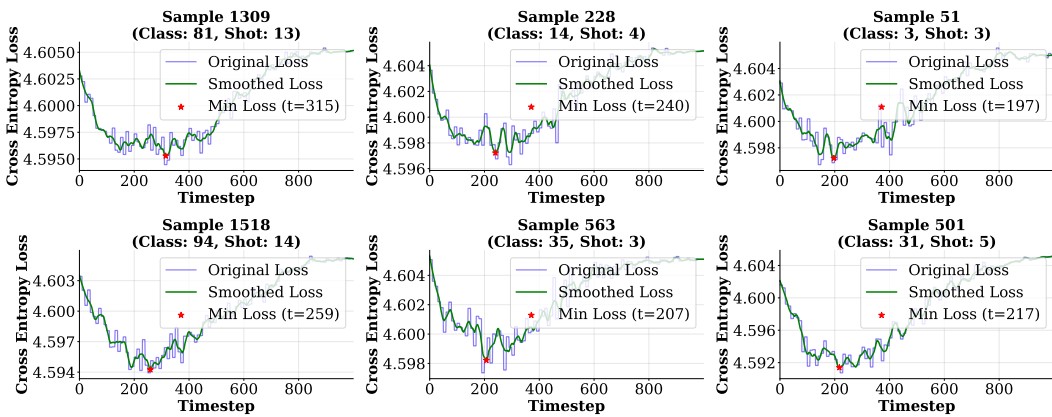

Figure 2: Relationship between time-step and discriminative ability (with cross-entropy loss as a negatively correlated metric): We randomly sampled 6 samples from the *FGVC-Aircraft* 16-shot support set (seed = 1) and plotted cross-entropy loss against time-step. Despite high curve fluctuation, the trend is clear: discriminative ability peaks within a specific time-step range.

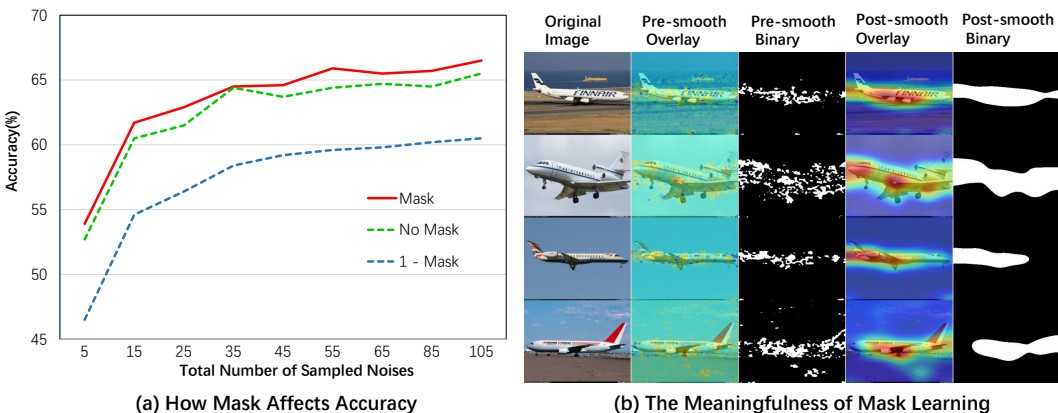

(a) How Mask Affects Accuracy     (b) The Meaningfulness of Mask Learning

Figure 3: Efficacy of mask learning mechanism. (a) Mask for each sample highlights evaluation of discriminative visual features, while "1 - Mask" highlights less discrimination-related visual features. (b) Learned Mask for each sample on the support set hightlights discriminative features, but is noisy without smoothing.

computed across the entire training set, significantly reducing the number of time-steps that need to be evaluated.

$$t^* = \arg\min_t \ F_t, t \in [0, 1]. \tag{6}$$

**Experimental Findings**. Our empirical results corroborate this hypothesis. We plotted the relationship between the discriminative ability $F_t$ and time-step t for numerous training samples in Figure 2. The samples showcased are sampled from the support set *FGVC-Aircraft* under the *16*-shot-*random seed* 1 setting. The plots consistently reveal that discriminative power peaks within a specific range of time-steps. Performance degrades at very earlier time-steps and at very late time-steps. This confirms the existence of an optimal sampling window and validates our strategy of concentrating inference within this high-gain, low-cost range.

## 4.2 NOISES

For a given test sample, the diffusion classifier selects the class that best reconstructs the noised image. However, this evaluation treats all pixels equally, conflating the reconstruction of the foreground object with that of the background environment. A class-specific LoRA (Hu et al., 2022) should excel at reconstructing the object of its class, while its ability to reconstruct the background is arbitrary and introduces randomness into the loss calculation. To mitigate this, we introduce a masking mechanism to focus the loss computation on salient object regions.

| | Result | FGVCAircraft(100 Classes) | | | | | VeRi-776(200 Classes) | | | | | PlantDiseases(38 Classes) | | | | |
|---|---|---|---|---|---|---|---|---|---|---|---|---|---|---|---|---|
| | | 1 | 2 | 4 | 8 | 16 | 1 | 2 | 4 | 8 | 16 | 1 | 2 | 4 | 8 | 16 |
| **Acc** | Without CCF | 48.5 | 56.4 | 64.9 | 75.6 | 81.0 | 38.9 | 57.8 | 75.2 | 88.4 | 95.6 | 50.9 | 61.9 | 76.2 | 86.5 | 92.1 |
| | With CCF | 52.5 | 59.2 | 66.8 | 75.2 | 80.8 | 41.4 | 60.4 | 75.7 | 88.3 | 95.4 | 57.2 | 68.9 | 80.6 | 88.8 | 93.2 |
| | Accuracy Variation | +4.0 | +2.8 | +1.9 | -0.4 | -0.2 | +2.5 | +2.6 | +0.5 | -0.1 | -0.2 | +6.3 | +7.0 | +4.4 | +2.3 | +1.1 |
| **Speed** | Speed Gain ($\times$) | 1.26 | 1.26 | 1.26 | 1.26 | 1.26 | 1.26 | 1.26 | 1.26 | 1.47 | 1.34 | 1.96 | 1.96 | 1.96 | 1.96 | 1.96 |
| **Class** | Candidate Number | 5 | 5 | 5 | 5 | 5 | 13 | 13 | 13 | 20 | 10 | 3 | 3 | 3 | 3 | 3 |

Table 1: Accuracy and Speed Variation with Class Candidate Filtering($CCF$ for short). Candidate Number refers to the number of classes left after filtering. Accuracy Variation and Speed Gain refer to the variation when applying $CCF$.

**Masking mechanism**. Our approach involves a three-step process to generate and apply these masks. First, we produce a dataset of $(\mathbf{x}, \mathbf{m})$ pairs from our training data. For each training sample $\mathbf{x}$, we generate an initial mask $\mathbf{m}$ by identifying regions in the reconstruction error map that are most informative for correct classification. This is done by minimizing cross entropy loss generated from the reweighted reconstruction maps under each class candidate. Second, we use this generated dataset to finetune a standard segmentation model as a mask generator, teaching it to predict the salient object regions directly from an input image. Third, during inference, we use this finetuned mask generator to generate a $\mathbf{m}$ for each test sample $\mathbf{x}$. This $\mathbf{m}$ is then applied to the reconstruction error map during inference, ensuring that our loss calculation is concentrated on the foreground object and less susceptible to background-induced noise.

**Experimental Findings**. The effectiveness of our masking mechanism is demonstrated in the accompanying Figure 3a. Results indicate that without this module, the classifier's performance degrades moderately. Given that marginal accuracy gains demand orders of magnitude more sampling, we conclude the masking mechanism boosts both efficiency and accuracy. Further, testing classification relying solely on unmasked regions led to a more pronounced performance drop, confirming our method's effectiveness. Quantitative results in Figure 3b validate the meaningfulness of learning masks: it enhances focus on visually essential regions for classification, e.g., object-containing areas. As the learned mask is noisy, we need to smooth it before using it as part of training set for segmentation model.

## 4.3 CLASSES

Exhaustive evaluation over an entire class vocabulary is computationally prohibitive and often redundant. A significant portion of candidates can be eliminated a priori using a more efficient method. We therefore introduce Class Candidate Filtering, a strategy designed to prune the search space of prospective classes before intensive analysis, thereby streamlining subsequent processes without compromising final accuracy.

**Class Candidate Filtering**. Our approach employs a highly efficient, pre-trained vision model, such as CLIP (Radford et al., 2021b), to perform this initial pruning. While CLIP's top-1 few-shot accuracy may not always match that of specialized classifiers, it exhibits excellent top-k accuracy, ensuring the ground-truth class is highly probable to be within its top predictions. We leverage this by using an adapted CLIP model to predict a small, high-probability subset of classes for a given sample, effectively filtering out the majority of irrelevant candidates at a minimal computational cost. A subsequent diffusion classifier then performs its evaluation exclusively on this reduced, pre-filtered set. This methodology substantially accelerates inference and synergistically integrates the robust discriminative power of the adapted CLIP model with the generative capabilities of the diffusion model.

**Experimental Findings**. We investigate the impact of class candidate filtering on accuracy and inference speed. Experiments are conducted on the *FGVCAircraft*, *Veri-776*, and *New Plant Diseases* datasets with random seed 1. Results are presented in Table 1. Our method filters out fewer than 10% of class candidates per sample, reducing the number of candidates for inference. However, this introduces prior errors stemming from CLIP's top-k accuracy and possibly conflicting discrimination principles, necessitating additional sampling for each retained candidate. Consequently, inference speed is not accelerated by more than 2x, yet the method still achieves speedup while preserving or boosting performance.

| Method & Shots | | FGVCAircraft | | | | | VeRi-776 | | | | | PlantDiseases | | | | |
|---|---|---|---|---|---|---|---|---|---|---|---|---|---|---|---|---|
| | | 1 | 2 | 4 | 8 | 16 | 1 | 2 | 4 | 8 | 16 | 1 | 2 | 4 | 8 | 16 |
| CLIP | Zero-Shot | | | 24.9 | | | | | | | | | | 9.1 | | |
| | CoOp | 22.8 | 28.4 | 32.4 | 37.7 | 40.5 | 11.4 | 13.9 | 18.1 | 30.3 | 34.1 | 36.3 | 55.2 | 68.1 | 76.6 | 85.3 |
| | Co-CoOp | 30.1 | 31.6 | 33.6 | 37.3 | 38.2 | 1.6 | 2.3 | 2.0 | 7.0 | 7.9 | 29.5 | 33.7 | 36.4 | 45.4 | 53.2 |
| | MaPLe | 30.1 | 33.0 | 33.8 | 39.4 | 40.7 | 35.2 | 40.7 | 44.4 | 57.5 | 68.1 | 20.0 | 34.6 | 34.4 | 46.9 | 67.5 |
| | AMU-Tuning | 10.6 | 11.0 | 13.1 | 15.5 | 16.4 | 17.3 | 26.6 | 39.9 | 56.0 | 70.0 | 25.2 | 33.9 | 42.5 | 50.7 | 57.5 |
| | SADA | 26.2 | 34.1 | 41.1 | 48.7 | 55.4 | 13.8 | 25.6 | 43.2 | 58.2 | 67.1 | 38.1 | 57.5 | 64.1 | 69.6 | 79.9 |
| OpenCLIP | Zero-Shot | | | 42.3 | | | | | | | | | | 18.8 | | |
| | Linear-probe | 18.4 | 32.5 | 44.1 | 55.0 | 59.8 | 18.4 | 32.5 | 44.1 | 55.0 | 59.8 | 39.5 | 41.4 | 58.1 | 66.3 | 71.0 |
| | Tip-Adapter | 47.7 | 51.6 | 54.7 | 58.4 | 62.2 | 47.7 | 51.6 | 54.7 | 58.4 | 62.2 | 50.9 | 61.9 | 71.7 | 80.2 | 80.1 |
| | Tip-Adapter-F | 48.4 | 53.9 | 56.9 | 62.0 | 67.4 | 48.4 | 53.9 | 56.9 | 62.0 | 67.4 | 49.4 | 50.0 | 59.2 | 77.3 | 88.0 |
| | Multi-modality | 42.3 | 54.0 | 63.2 | 70.0 | 74.3 | 45.7 | 65.3 | 76.5 | 84.9 | 89.3 | 63.9 | 78.1 | 86.9 | 92.4 | 95.1 |
| DM | Zero-Shot | | | 24.3 | | | | | | | | | | 6.1 | | |
| | Full TiF | 48.5 | 55.8 | 64.2 | 74.2 | 79.9 | 41.9 | 60.7 | 78.2 | 91.2 | 96.8 | | | | | |
| | Degraded TiF | 42.1 | 51.7 | 61.3 | 70.5 | 75.8 | 38.9 | 57.8 | 75.2 | 88.4 | 95.6 | 33.9 | 47.2 | 61.7 | 76.8 | 86.5 |
| | Ours | 52.9 | 59.4 | 67.1 | 74.8 | 81.1 | 40.3 | 57.6 | 74.5 | 87.5 | 95.4 | 57.5 | 68.6 | 80.6 | 87.6 | 92.4 |
| Speed | Degraded TiF(×) | 10.0 | 10.0 | 10.0 | 10.0 | 10.0 | 21.1 | 21.1 | 21.1 | 21.1 | 21.1 | | | | | |
| | Ours(×) | 12.6 | 12.6 | 12.6 | 12.6 | 12.6 | 28.4 | 28.4 | 28.4 | 39.7 | 31.8 | | | | | |

Table 2: *N*-shot Accuracy on *FGVCAircraft*, *Veri-776* and *New-plant-diseases*. Zero-shot experiment for the re-identification dataset *VeRi-776* is not conducted as the class names(like "car 0", "car 1") are meaningless for zero-shot multi-modal models. *Full TiF* refers to *TiF* with number of sampled noises under the original setting, while *Degraded TiF* samples less noises. We did not conduct *Full TiF* on *New Plant Diseases* as it is unacceptably slow(more than 12 days for a single shot and single seed). The speed is evaluated with*Full TiF* speed regarded as 1. Our method samples slightly less noises than *Degraded TiF* while maintaining or even improving the original performance. Linear Probe refers to Radford et al. (2021a). The speed gain is evaluated with the metric "sampled noises"

## 5 EXPERIMENTS

### 5.1 SETTINGS

**Datasets**. We conduct our experiments on three fine-grained and customized datasets, which covers diverse scenarios. We used: (1) *FGVCAircraft* (Maji et al., 2013) comprises aircraft images across 100 categories, where the visual differences between these categories are subtle. (2) *VeRi-776* (Liu et al., 2016a;b) features 200 vehicle IDs captured by 20 cameras, offering a wide variety of viewing angles and environmental scenarios. (3) *New Plant Diseases* (Lee et al., 2020) consists of healthy and diseased crop leaves which is categorized into 38 different classes.

**Evaluation Details**. We evaluate on *K*-way-*N*-shot few-shot learning setting. The experimental setup is challenging as the range of our K value (38 to 200) is significantly broader, in contrast to the constrained $K = 5$ that is commonly used in traditional $FSL$ settings. On *FGVCAircraft*, *New Plant Diseases* which does fine-grained classification, we sampled the few-shot set from its train split and evaluated accuracy on its test split. On *Veri-776* which does customized re-identification task, we select 200 individual vehicles and build its train and test dataset. As these datasets do not suffer severely from class imbalance, the metric we've adopted is top-1 accuracy.

**Implementation Details**. We inherit settings of TiF (Yue et al., 2024) in finetuning and prompt forming. To be more specific, we utilized SD 2.0 as our diffusion backbone. Regarding the rank of LoRA matrices, we set it to 16 for all datasets. A fixed rare token identifier, [V] = "hta", was employed across all experiments following dreambooth. The formed prompt is "a photo of [V] [C], a type of [SC]", where [C] denotes selective class name and [SC] is a dataset-specific super-class name, i.e., "aircraft" on *FGVCAircraft* and "plant disease" on *New Plant Diseases*. For inference, we've inherited maximum confidence filtering (See Appendix).

We adopt time-step learning, mask-learning and class candidate filtering for best acceleration and performance. For time-step learning, 100 $(t, \epsilon)$ pairs are sampled for each training sample to evaluate the best time-step $t^*$, and time-step range is set to be not longer than 300 and $2 \times t^*$. For mask learning, mask is learned on reconstruction error map meaned across 16 or 32 sampled $(t, \epsilon)$ pairs sampled from the golden time-step range generated from time-step learning. This process is done by first use mask-reweighted reconstruction error map of each class candidate to build cross entropy loss, then minimize the loss with L2 regularization. The mask is then post-processed via average pooling and min-max normalized. We train the mask generator with pretrained segmentation model

| | FGVCAircraft | | | | | VeRi-776 | | | | | PlantDiseases | | | | |
|---|---|---|---|---|---|---|---|---|---|---|---|---|---|---|---|
| | 1 | 2 | 4 | 8 | 16 | 1 | 2 | 4 | 8 | 16 | 1 | 2 | 4 | 8 | 16 |
| CLIP Acc | 41.1 | 53.4 | 62.5 | 69.1 | 74.0 | 36.4 | 65.5 | 77.2 | 85.0 | 89.7 | 57.6 | 78.3 | 86.0 | 93.0 | 94.5 |
| Diffusion Acc | 53.6 | 59.5 | 67.1 | 75.1 | 81.0 | 41.4 | 60.4 | 77.4 | 87.2 | 95.4 | 62.6 | 70.3 | 80.6 | 88.9 | 93.2 |
| Integration Acc | **55.2** | **61.2** | **68.1** | **75.6** | **81.2** | **49.2** | **71.2** | **80.6** | **90.1** | **95.6** | **67.3** | **81.9** | **87.6** | **93.2** | **95.6** |
| Acc Gain | 1.6 | 1.7 | 1.0 | 0.5 | 0.2 | 8.0 | 5.7 | 3.2 | 2.9 | 0.2 | 4.7 | 3.6 | 1.6 | 0.2 | 1.1 |
| Threshold | 0.5 | 0.5 | 0.5 | 0.5 | 1.2 | 0.1 | 0.1 | 0.1 | 0.1 | 0.1 | 0.5 | 0.5 | 0.5 | 0.5 | 0.5 |

Table 3: CLIP denotes SOTA CLIP-based multi-modal methods, while Diffusion refers to our method. CLIP confidence is the top-two logit difference, with a threshold separating high/low confidence samples. The integrated framework uses CLIP for high-confidence samples and Diffusion otherwise.

*FCN-ResNet50* network. When doing inference, we utilize *Tip-Adaptor* or *Tip-Adaptor-F* adapted to *ViT-H-14* backbone for class candidate filtering, and relearn time-step with *filtered class candidates*. The reconstruction error map is reweighted by the class-specific generated mask for better evaluation.

**Baselines**. We compared with three types of methods mentioned in *Related Work*: 1) TipAdapter based on Equation4. 2) Prompt tuning methods CoOp, Co-CoOp and MaPLe, which aim to learn $y_c$ in Equation 4. 3) Multiple information methods Multi-modality, SADA, and AMU-Tuning. We've adopted two CLIP variants: *ViT-B-16* backbone for CLIP and *ViT-H-14* backbone for OpenCLIP. The former was trained on 400 million image-caption pairs, while the latter is trained on the *LAION-2B*. It is expected that all methods are implemented on better-performing backbone *ViT-H-14*. However, some methods are not implemented on OpenCLIP because their official implementations are specifically designed for CLIP and difficult to be fairly reproduced on OpenCLIP. We also compared with the zero-shot Diffusion Classifier and TiF, which is based on SD. For TiF comparison, we conduct experiment with two settings: Numbers of sampled noises following the original TiF setting(referred to as Full TiF), and degraded numbers of sampled noises(referred to as Degraded TiF) for fairer comparison with FastTiF(Our method).

## 5.2 MAIN RESULTS

**Overall Results**. As shown in Table 2, our method outperforms most CLIP-based methods across three benchmark datasets—*FGVCAircraft* (fine-grained classification), *VeRi-776* (re-identification), and *New Plant Diseases* (fine-grained classification). On *FGVCAircraft*, it outperforms all compared CLIP and OpenCLIP variants across all shot settings: 4.5% lead for 1-shot and 6.8% lead for 16-shot. For *VeRi-776* and *New Plant Diseases*, it shows competitive performance: on *VeRi-776*, its 16-shot accuracy surpasses most CLIP-based baselines; on *New Plant Diseases*, it reaches **92.4%** at 16 shots, trailing only one OpenCLIP variant.

Our work maintains performance while accelerating Diffusion Classifier. Our method exceeds Degraded TiF by slight acceleration(1.26× 1.96×) with better performace(at least 4% lead on *FGVC-Aircraft* and *New Plant Diseases*, only less then 1% drop on hight shots of *VeRi-776* but with more acceleration). When compared with Full TiF, it accelerates the method vastly, shortening the minute-wise inference per sample to second-wise(approximately 3s per sample for *FGVC-Aircraft* and *New Plant Diseases* and 11s for *Veri-776*). For testing accuracy, it exceeds *FGVC-Aircraft* on all given shots, and degrades by at most 3.7% on *Veri-776*.

Though our method does not exceed the SOTA method Multi-modality on *Veri-776* and *New Plant Diseases* on most shots, we've exhibited extensive experiments to integrate our method and Multi-modality, which exceeds both methods. The result is shown in ABLATIONS.

## 5.3 ABLATIONS

**Relationship between CLIP and Diffusion Classifier**. We stated previously CLIP-based methods can assist diffusion-based methods in class filtering. The following illustrates how diffusion-based methods can alleviate CLIP's classification ambiguity.

The adopted CLIP-based method is the SOTA Multi-modality variant under the CLIP framework, which employs a linear head attached to a fine-tuned visual encoder for classification. As plotted in

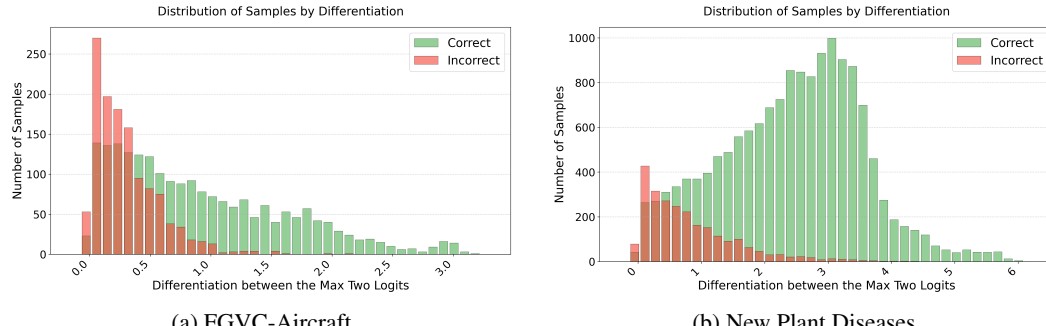

(a) FGVC-Aircraft  (b) New Plant Diseases

Figure 4: Relationship between CLIP Confusion and Classification Correctness. Experiments were conducted on two datasets with 4 shots and a random seed of 1. The CLIP method we adopt is the SOTA CLIP-based method Multi-modal, which selects the target class via maximum logit. CLIP confidence is defined as the difference between the top-two logits. If the confidence is low, it will be more likely to misclassifiy.

| shots | FGVCAircraft | | VeRi-776 | | New Plant Diseases | |
|---|---|---|---|---|---|---|
| | Learning | TiF Estimation | Learning | TiF Estimation | Learning | TiF Estimation |
| 1 | 52.3/284 | 52.7/254 | 37.0/264 | 36.2/238 | 57.2/227 | 57.1/307 |
| 2 | 59.5/303 | 59.8/255 | 52.9/278 | 51.9/236 | 68.9/225 | 70.4/284 |
| 4 | 67.1/302 | 67.0/254 | 74.0/267 | 72.9/237 | 80.6/231 | 80.6/278 |
| 8 | 75.1/247 | 75.1/244 | 86.4/271 | 85.6/236 | 88.8/184 | 89.1/281 |

Table 4: Time-step Learning(Ours) vs. TiF Ratio Time-step Estimation(Ablation). The recorded data is formed in *Accuracy/ Estimated Time-step*. Both methods sample time-steps around the estimated best time-step. The former method learns this time-step, while that latter choose the time-step with the largest TiF weight.

Figure 4, if the difference between the top two logits is low, the method tend to misclassify. Thus we can define CLIP classification confidence with this metric.

To mitigate CLIP's ambiguity, we propose a complementary strategy: CLIP-based predictions are adopted if the logit difference exceeds a threshold; otherwise, the diffusion classifier serves as an alternative. Evaluations on *FGVC-Aircraft*, *Veri-776*, and *New Plant Diseases* (seed=1) show that the integrated method achieves slightly higher test accuracy than individual methods (Table 3), verifying the non-contradictory and complementary nature of the two classification frameworks.

**The Effectiveness of Time-step Learning**. Time-step Learning propose to sample in a "golden range" around the learned optimal time-step. However, this time-step can be attained via other sources. The TiF arbitrary-given ratio gives higher weight of reconstruction error evaluation at a certain time-step, and this time-step can be regarded as the optimal time-step as well. We investigate how our learning method distinguishes from TiF time-step evaluation.

Table 4 presents the experimental results, comparing classification accuracy and the estimated optimal time-step between two strategies. Across all datasets and shot configurations, the accuracy values are highly consistent, indicating that the specific choice of time-step has a negligible impact on performance. The variance of the learned optimal time-step is significantly larger than that of the TiF-estimated time-step, which is possibly caused by the fluctuation of the curve(Figure 2). We thus conclude that TiF-estimated time-step can be a better alternative for our method. However, our method still has analytical values: Only our method can evaluate quantize the relationship between time-step and discrimination ability, and illustrate the existence of "golden time-step range".

## 6  CONCLUSIONS

We present FastTiF to address the computational inefficiency of diffusion classifier via resolving its three bottlenecks: extensive time-step sampling, uniform noise evaluation, and redundant class candidate evaluation. We've discussed the effect of key components to resolve these problems: Why they are plausible or how they contribute to acceleration and performance. Extensive results show that our method greatly accelerates the original work while maintaining or even improving the performance. We conclude that our method outperforms most CLIP-based methods, and is possible for integration with CLIP-based framework.

**Ethics Statement**. This study does not involve any ethical concerns and carries no ethical risks whatsoever.

**Reproducibility Statement**. To guarantee the reproducibility of the proposed method, we have taken the following measures: (1) Both our fine-tuning and inference codes, as well as the implementation details, will be publicly accessible. (2) Detailed settings (dataset splits, hyperparameters, environments) are in the appendix. (3) Benchmark datasets are publicly accessible with clear pre-processing notes,

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

# A APPENDIX

## A.1 LLM USAGE DECLARATION

Large Language Models (LLMs) were leveraged ONLY to support manuscript writing and polishing. Specifically, an LLM was employed to help refine language, boost readability, and ensure clarity across different paper sections. It assisted with tasks like sentence rephrasing, grammar checking, and enhancing the text's overall flow.

Notably, the LLM had no role in ideation, research methodology, or experimental design. All research concepts, ideas, and analyses originated from and were executed by the authors. The LLM's contribution was strictly confined to elevating the paper's linguistic quality, with no involvement in scientific content or data analysis.

The authors **fully assume responsibility** for the manuscript's content, including text generated or polished by the LLM. We've ensured LLM-produced text complies with ethical guidelines, avoiding plagiarism or scientific misconduct.

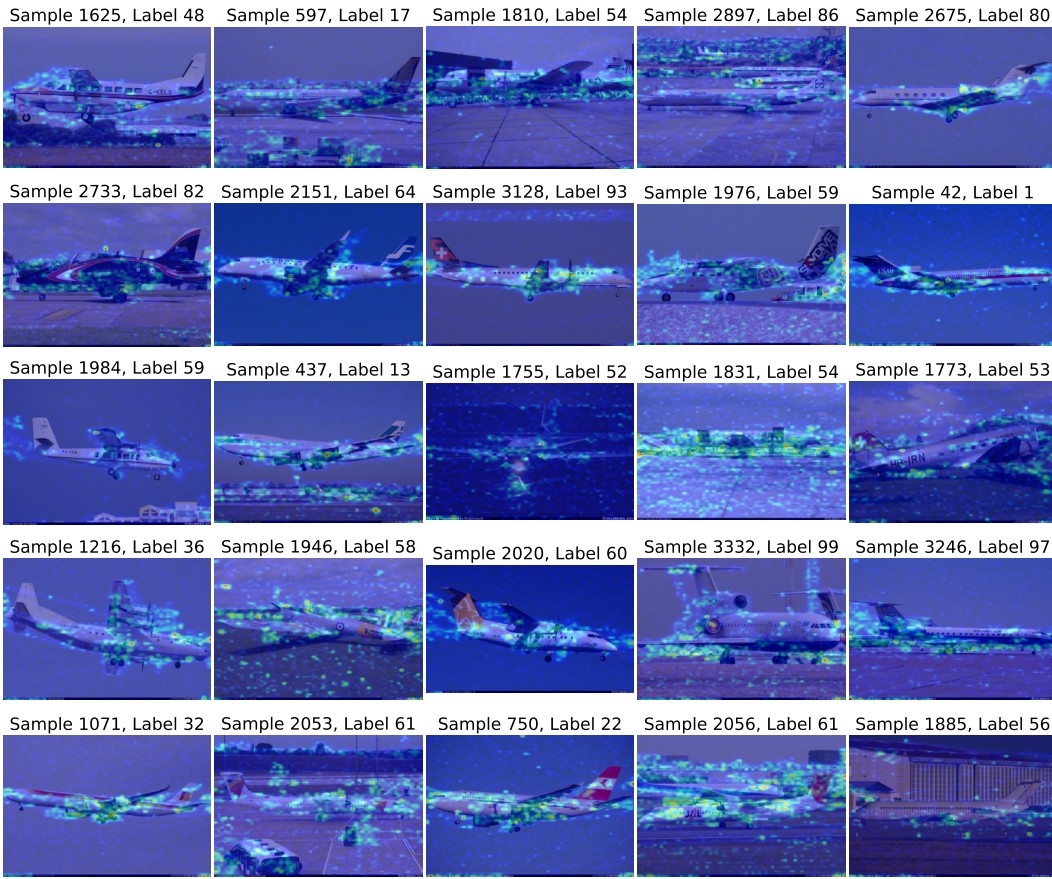

Figure 5: Noise of Reconstruction Error in Regions with Weak Discriminative Features. The images presented herein are randomly sampled from the test set of *FGVC-Aircraft*. Each visualization overlays the original image with a reconstruction error heatmap, where warmer color tones (transitioning from green to red) indicate higher magnitudes of reconstruction error.

## A.2 MAXIMUM CONDIDENCE FILTERING

We've inherited maximum confidence filtering proposed in Li et al. (2023), which does inference with multi-stage for more efficient inference. Suppose we sample for each class candidate N $(t, \epsilon)$ pairs and do N trials, the total sampled number is $C \times N$. A better multi-stage testing method

can address this issue, and its implementation is as follows: the evaluation process is divided into multiple stages. In each stage, a certain number of trials are conducted for each remaining class, and then the classes with the highest average error are eliminated, thereby allocating more computing resources to reasonable candidate classes. For example, on *Pets* dataset of Li et al. (2023)'s setting, N stages is set to 2. In the first stage, 25 trials are conducted for each class, and the 5 classes with the smallest average error are retained through pruning. In the second stage, an additional 225 trials are conducted for these 5 classes, and only one class candidate is left as the target class. It is much faster for doing 250 trials for each class candidate.

### A.3 MASK LEARNING

For better understanding and easier expression, we see discriminative area as the object area, and the less discriminative area as background.

#### A.3.1 MOTIVATION FOR MASK LEARNING

As plotted in 5, Reconstruction error may concentrate in discriminative regions. However, it should be noted that in areas with fewer discriminative features (e.g., the background), reconstruction error noise persists. Without randomness, the reconstruction error in the background ought to remain consistent. This illustrates our insight that the diffusion classifier's ability to reconstruct the background is arbitrary and introduces randomness into loss calculation.

We further provide a possible explanation for this. Suppose we have an image of the aircraft "707-320" and use class prompts for two other classes: "770-320" and "A330-200". The object reconstruction process varies with different class-conditional guidance, but this is not true for background reconstruction. Class prompts do not contain environmental information, so the evaluation of reconstruction error in the background is entirely influenced by randomness. It is therefore wise to downweight the evaluation of this area.

#### A.3.2 EXPLANATION FOR HOW MASK LEARNING WORKS

We've stated that in experiment, if we learn a mask for a certain sample on the reconstruction error map, the map will highlight visually discriminative area. But how is it possible? We give out an explanation. Ideally speaking, for a single sample, class conditional guidance given by class candidates vary on object reconstruction, and remains consistent on background area. So if learned, the mask does highlight the object area.

## A.4 TABLE OF SYMBOLS

## SYMBOL TABLE FOR "HOW TO TRAIN YOUR DIFFUSION MODELS FOR FEW-SHOT CLASSIFICATION"

| Symbol | Description | Symbol | Description |
|---|---|---|---|
| $DMs$ | Diffusion Models: Generative models used for few-shot classification by predicting noise added to images during diffusion. | $\hat{c}$ | Predicted class label for input image $x$, determined by minimizing the expected reconstruction error over noises and time-steps. |
| $E(x,t,c)$ | Reconstruction error map for image $x$ at time-step $t$ under class condition $c$, defined as $\text{Square}(DM(tx+(1-t)\epsilon,t,c)-\epsilon) \in \mathbb{R}^{Ch \times H \times W}$ (where $\epsilon$ is sampled noise). | $\epsilon$ | Random noise sampled from a standard normal distribution $\mathcal{N}(0,I)$, added to images during the diffusion noising process. |
| $t$ | Diffusion time-step: Ranges over $U(0,1)$ for flow matching or $[0,1000]$ for DDPM. | $t^*$ | Optimal time-step for classification, minimizing the cross-entropy loss $F_t$ (negatively correlated with discriminative ability). |
| $K$ | Number of classes in the $K$-way few-shot classification task. | $N$ | Number of labeled samples (shots) per class in the support set of $N$-shot learning. |
| $S$ | Support set: Defined as $\{(x_i,y_i)\}_{i=1}^{K \times N}$, containing $K \times N$ labeled samples to define novel classes. | $Q$ | Query set: Unlabeled samples from the same $K$ classes as $S$, used to evaluate model generalization. |
| $V(x)$ | Output embedding of image $x$ from CLIP's visual encoder. | $T(y_c)$ | Output embedding of class text prompt $y_c$ from CLIP's text encoder. |
| $\text{CLIP}(x,y_c)$ | Cosine similarity between image embedding $V(x)$ and text embedding $T(y_c)$, i.e., $\cos(V(x), T(y_c))$. | $\theta_c$ | Class-specific LoRA (Low-Rank Adaptation) parameters: Fine-tuned via DreamBooth to customize diffusion models for class $c$. |
| $DM_{\theta_c}$ | Class-conditional diffusion model: Specialized for class $c$ via LoRA parameters $\theta_c$. | $r_t$ | Time-step ratio in the TiF framework: Used to debias classification and eliminate spurious correlations. |
| $L_t(x,c)$ | Expected reconstruction error for image $x$, class $c$ at time-step $t$, defined as $\mathbb{E}_{\epsilon \sim \mathcal{N}(0,I)}\|E(x,t,c)\|$. | $F_t$ | Cross-entropy loss over negative reconstruction errors $-L_t(x,c)$: Negatively correlated with the classifier's discriminative ability at time-step $t$. |
| $m$ | Spatial mask: Generated by a fine-tuned segmentation model to down-weight background regions and focus on salient object features in error calculation. | $C$ | Full set of candidate classes in the few-shot classification task. |
| $C_R$ | Reduced candidate class set: Obtained by filtering $C$ via CLIP (prunes irrelevant classes to accelerate inference). | FGVCAircraft | Fine-grained benchmark dataset: Contains 100 aircraft categories with subtle visual differences. |
| VeRi-776 | Vehicle re-identification dataset: Features 200 vehicle IDs captured across 20 cameras. | New Plant Diseases | Fine-grained dataset: Includes 38 classes of healthy/diseased crop leaves. |
| Full TiF | TiF framework with the original number of sampled noises (used as a baseline). | Degraded TiF | TiF framework with fewer sampled noises (used for fair speed/accuracy comparison with FastTiF). |
| CCF | Class Candidate Filtering: Strategy to prune irrelevant class candidates via CLIP, reducing inference computation. | SD 2.0 | Stable Diffusion 2.0: Used as the diffusion backbone for FastTiF. |
| $[V]$ | Fixed rare token identifier (set to "hta") in DreamBooth prompts for class-specific fine-tuning. | $[C]$ | Selective class name in the prompt template ("a photo of [V] [C], a type of [SC]"). |
| $[SC]$ | Dataset-specific super-class name in prompts (e.g., "aircraft" for FGVCAircraft, "plant disease" for New Plant Diseases). | FCN-ResNet50 | Pre-trained segmentation model: Fine-tuned as the mask generator for FastTiF's mask learning module. |

