# OpenReview forum: "HOW TO TRAIN YOUR DIFFUSION MODELS FOR FEW- SHOT CLASSIFICATION"
_ICLR.cc/2026/Conference — ICLR 2026 Conference Withdrawn Submission_

### Official Review · Reviewer_NtX7 · 2025-10-24

**Soundness:** 3
**Presentation:** 1
**Contribution:** 2
**Rating:** 4
**Confidence:** 3

**Summary:**

The manuscript proposes FastTiF, an incremental method improved from TiF [1] to accelerate diffusion-based few-shot classification  while maintaining or improving accuracy. The authors address three computational bottlenecks: (1) extensive time-step sampling through time-step learning, (2) uniform noise evaluation through mask learning, and (3) large candidate class pools through CLIP-based filtering. Experiments on three datasets (FGVCAircraft, VeRi-776, New Plant Diseases) show 10-40 time speedups with competitive accuracy compared to TiF baseline and other CLIP-based fine-tuning methods.

[1] Few-shot Learner Parameterization by Diffusion Time-steps, CVPR 2024

**Strengths:**

1. The paper is well-motivated with good empirical support. The analysis experiments, especially Figure 2, effectively demonstrate the existence of a "golden time-step range" where discriminative ability peaks, making the motivation very clear.

2. The approach addresses the main computational bottlenecks from 3 perspectives (time-steps, noise evaluation, and class candidates) with clear rationale for each component.

3. The baseline selection seems to be adequate, comparing against diverse CLIP-based methods (prompt tuning, adapter-based, multi-modal) and diffusion-based approaches. Achieving 10-40× acceleration is a good practical contribution that could enable real-world deployment.

**Weaknesses:**

1. The writing is imprecise and unpolished throughout. There are numerous typos: "eg.", "dreambooth", "Tip-Adaptor", "performace", "hight". The formatting is inconsistent with parentheses spacing and dataset names (PlantDiseases vs New Plant Diseases). Some parts are written unclearly, especially in the Experiments and hard to understand the settings such as "The adopted CLIP-based method is the SOTA Multi-modality variant under the CLIP framework" and the "Degraded TiF" baseline terminology is confusing because it was not described in detail. Why not report the number of noise sample counts directly for this baseline?

2. Several experimental details are missing. Table 2 omits zero-shot results for VeRi-776 when provided for other datasets. The paper tests on different datasets than the original TiF paper, missing comparisons on ISIC2019 and DukeMTMC-reID datasets. Why don't compare on these datasets for a fair comparison?  How sensitive are results to mask learning hyperparameters and architecture choices?

3. Limited Novelty. The works seems to be an incremental improvement of TiF rather than a fundamental methodological advance. The CLIP filtering is a well-known technique. The paper's own ablation (Table 4) reveals that time-step learning achieves nearly identical results to TiF's existing ratio-based time-step estimation, directly undermining one of three claimed contributions. The authors acknowledge this but don't adequately address why it should be considered a novel contribution.

4. The limitation of proposed method was not discussed properly. (see Questions for more detail).

**Questions:**

1. A failure case for FastTiF is degraded performance on higher shots in Table 1. Could the authors provide an explanation for this phenomenon? Although I agree that CLIP filtering usually yield satisfactory top-k score, utilizing CLIP score may lead to false positives and could hurt performance [2]. Could the authors validate whether it is the case in this scenario?
2. Given Table 4 shows TiF's time-step estimation works equally well, the need of learning time-step is not convincing. Could the authors clarify more on this?
3. Why is Full TiF "super slow" on Plant Diseases (38 classes) but computable for FGVCAircraft (100 classes) or VeRi-776 (200 classes)?
4. How many noise samples does Degraded TiF use compared to Full TiF and your method? Why do you need to introduce this intermediate baseline?
5. What is the training time overhead for mask learning, and when does this complexity pay off compared to simply using more noise samples?

[2] Sieve: Multimodal dataset pruning using image captioning model, CVPR 2024

---

### Official Review · Reviewer_nZMd · 2025-10-30

**Soundness:** 3
**Presentation:** 3
**Contribution:** 2
**Rating:** 4
**Confidence:** 3

**Summary:**

This paper introduces FastTiF, a fast and accurate diffusion-based few-shot classification method. FastTiF accelerates standard diffusion classifiers through three key steps: (1) time-step learning, which focuses inference on an optimal “golden” diffusion range; (2) mask learning, which guides the model to compute reconstruction loss only on salient object regions; and (3) class candidate filtering, which leverages CLIP to preselect a small set of likely classes.Together, these steps accelerate diffusion-based classifiers by over an order of magnitude, while maintaining or even improving classification accuracy.

**Strengths:**

Complementary integration: The paper effectively combines the strengths of diffusion and CLIP-based models, showing that they are complementary.

Significant acceleration: FastTiF achieves up to an order-of-magnitude speedup over TiF while maintaining or even improving accuracy.
Time-step optimization: The proposed time-step learning efficiently restricts sampling to a golden range,reducing unnecessary computations.

Mask learning mechanism: The introduction of class-object masks helps focus on salient regions, improving both efficiency and robustness.

**Weaknesses:**

Limited benchmarks: The evaluation is restricted to only three datasets, lacking experiments on broader or more standard FSL benchmarks such as miniImageNet or CUB.

Incomplete comparison: The paper does not provide a comprehensive comparison with conventional FSL approaches or other foundation-model-based FSL methods.

Shallow integration analysis: The integration between CLIP and diffusion models is based mainly on confidence thresholds, without a deeper theoretical or systematic analysis of when each model performs better.

Incremental contribution:The proposed improvements (time-step selection, mask learning, class filtering) are mainly engineering refinements rather than novel theoretical contributions.

**Questions:**

Most prior FSL works report results on miniImageNet, CUB, and tieredImageNet under 5-way 1-shot and 5-way 5-shot settings. Could you evaluate FastTiF on these standard benchmarks to enable a fair comparison with conventional FSL methods?

Have you considered testing cross-domain transfer, for example, training on miniImageNet and evaluating on CUB or tieredImageNet, to assess the model’s generalization capability?

The CLIP–Diffusion integration is based solely on a confidence threshold. Have you explored more adaptive or probabilistic fusion strategies that might improve performance further?

The current comparisons mostly focus on CLIP-based methods. Could you include more conventional FSL baselines (e.g., ProtoNet, MatchingNet,MSENet  or Meta-Baseline) to clarify where FastTiF stands relative to traditional paradigms?

It would be valuable to analyze failure cases, especially where the integration misclassifies samples. For instance, were these cases correctly handled by CLIP or Diffusion individually, and does this suggest that the confidence threshold might not always capture uncertainty effectively?

---

### Official Review · Reviewer_PEW7 · 2025-10-30

**Soundness:** 2
**Presentation:** 3
**Contribution:** 3
**Rating:** 4
**Confidence:** 3

**Summary:**

This paper proposes FastTiF, aiming to improve the inference efficiency and accuracy of Diffusion Models (DMs) in Few-shot Classification (FSL) tasks. The authors argue that generative diffusion models possess complementary potential to discriminative models (such as CLIP) in few-shot settings; however, existing diffusion classifiers suffer from extremely high inference costs, as they require averaging reconstruction errors across all classes, multiple noise samples, and time steps. To address this, the paper introduces three core improvements: Time-step Learning; Pixel-level Saliency Mask Learning; Class Candidate Filtering (CCF). Furthermore, the paper explores the complementarity between CLIP and Diffusion Classifiers, and proposes a hybrid inference strategy to further enhance performance.

**Strengths:**

1.	The paper is clearly written, well-structured, and effectively supported with figures and tables.
2.	It identifies three computational bottlenecks of diffusion classifiers and designs corresponding improvement methods for each.

**Weaknesses:**

1.	The literature review is not comprehensive enough, with insufficient references to recent related works.
2.	The authors directly present the mathematical formulation of the diffusion classifier without explaining how these equations relate to the stated problems (low efficiency and high computational cost), nor do they explicitly clarify why the method is computationally expensive.
3.	Although the proposed time-step learning is empirically supported, it lacks theoretical analysis regarding how discriminative ability changes with diffusion steps.
4.	The ablation studies are not intuitive; Section 5.3 does not systematically analyze the effect of each proposed module or present clear ablation results.
5.	The complementarity of the CLIP + Diffusion hybrid framework is not sufficiently convincing — the integration strategy is merely based on a logit-difference threshold, without prior analysis of the performance gap or complementarity mechanism between the two models.

**Questions:**

Please see the weakness.

---

### Official Review · Reviewer_hCY8 · 2025-11-01

**Soundness:** 3
**Presentation:** 3
**Contribution:** 3
**Rating:** 4
**Confidence:** 3

**Summary:**

This paper proposes FastTiF, a method that improves both the efficiency and accuracy of diffusion classifiers for few-shot classification. The authors introduce three mechanisms: (a) time-step learning to identify the optimal sampling window during inference; (b) mask learning to focus on foreground object regions while suppressing background reconstruction-error noise; and (c) candidate label selection using a CLIP‐based method to reduce the set of potential classes.  Experiments demonstrate that FastTiF outperforms baselines in most cases. They also considers combining FastTiF and CLIP-based methods.

**Strengths:**

1. The paper presents a clear and well-motivated problem setup, proposing three reasonable modules: time-step learning, mask learning, and class candidate filtering-that intuitively address distinct sources of inefficiency in diffusion-based few-shot classification.
2. The experiments demonstrate the effectiveness of the proposed model.

**Weaknesses:**

1. The discussion of prior work on accelerating diffusion-based classifiers is too brief.
2. The explanation of the proposed modules lacks sufficient clarity and detail.
3. The training cost of the proposed method is not discussed. It is unclear how the speedup is measured and whether the comparison is fair.

**Questions:**

1. Do the three modules require retraining for each dataset or task, or are they trained once and for all? If they need to be trained for each task, how does that fit in the the few-shot learning scenario? How many samples are required and how much time? Will the comparisons in the evaluation be unfair?

2. Lines 319–323 state that the method “filters out fewer than 10% of class candidates per sample,” while Table 1 shows that only 5–10% of the classes remain (e.g., 5 out of 100 for FGVCAircraft), implying that more than 90% are removed. How many classes are actually removed in practice? Moreover, how are the retained candidates defined and how is the additional sampling performed (described in lines 320-322)?

3. Lines 357-359: Larger K makes the problem more challenging but may also give the proposed method more advantage. How do the methods compare when K is small?

4. What samples are used to train the proposed modules in the experiments? Is there information leakage here, making it not a true few-shot learning scenario?

---

### Note · Authors · 2025-11-12

I have read and agree with the venue's withdrawal policy on behalf of myself and my co-authors.